# Impaired hematopoiesis and embryonic lethality at midgestation of mice lacking both lipid transfer proteins VPS13A and VPS13C

Peng Xu[1,2,3,4,5]*, Rubia Isler Mancuso[6,7], Marianna Leonzino[1,2,3,4,5¤], Caroline J. Zeiss[8], Diane S. Krause[6,7,9], Pietro De Camilli[1,2,3,4,5]*

1 Department of Neuroscience, Yale University School of Medicine, New Haven, Connecticut, United States of America, 2 Department of Cell Biology, Yale University School of Medicine, New Haven, Connecticut, United States of America, 3 Program in Cellular Neuroscience, Neurodegeneration and Repair, Yale University School of Medicine, New Haven, Connecticut, United States of America, 4 Aligning Science Across Parkinson's (ASAP) Collaborative Research Network, Chevy Chase, Maryland, United States of America, 5 HHMI, Yale University School of Medicine, New Haven, Connecticut, United States of America, 6 Department of Laboratory Medicine, Yale University School of Medicine, New Haven, Connecticut, United States of America, 7 Yale Stem Cell Center, Yale University School of Medicine, New Haven, Connecticut, United States of America, 8 Department of Comparative Medicine, Yale School of Medicine, New Haven, Connecticut, United States of America, 9 Department of Pathology, Yale School of Medicine, New Haven, Connecticut, United States of America

¤ Current address: Institute of Neuroscience, Consiglio Nazionale delle Ricerche, Rozzano, Milan, Italy
* peng.xu@yale.edu (PX); pietro.decamilli@yale.edu (PDC)

## Abstract

VPS13 is the founding member of a family of proteins that mediate lipid transfer at intracellular membrane contact sites by a bridge-like mechanism. Mammalian genomes comprise 4 *VPS13* genes encoding proteins with distinct localizations and function. The gene duplication resulting in *VPS13A* and *VPS13C* is the most recent in evolution and, accordingly, these two proteins are the most similar to each other. However, they have distinct subcellular localizations and their loss of function mutations in humans are compatible with life but result in two different age-dependent neurodegenerative diseases, chorea-acanthocytosis and Parkinson's disease, respectively. Thus, it remains unclear whether these two proteins have overlapping functions. Here, we show that while *Vps13a* KO and *Vps13c* KO mice are viable, embryonic development of *Vps13a*/*Vps13c* double knockout (DKO) mice is arrested at midgestation. Prior to death, DKO embryos were smaller than controls, were anemic and had a smaller liver, most likely reflecting defective embryonic erythropoiesis which at this developmental stage occurs primarily in this organ. Further analyses of erythroid precursor cells showed that their differentiation was impaired and that this defect was accompanied by activation of innate immunity as revealed by upregulation of interferon stimulated genes (ISGs). Additionally, the RIG-I and MDA5 components of dsRNA triggered innate immunity were found upregulated in the DKO fetal liver. Activation of innate immunity may result from loss of integrity of the membranes

**Data availability statement:** All RNAseq files are available from the GEO database (accession number GSE297002). Other data, code, protocols, and key lab materials used and generated in this study are listed in a Key Resource Table alongside their persistent identifiers at Zenodo DOI: https://doi.org/10.5281/zenodo.15375803.

**Funding:** This work was supported in part by grants from the NIH https://www.nih.gov/ (NS36251 and DA018343) and by the Aligning Science Across Parkinson's grants https://parkinsonsroadmap.org/ through the Michael J. Fox Foundation for Parkinson's Research (ASAP-000580) to PDC and the Yale Cooperative Center of Excellence in Hematology https://medicine.yale.edu/labmed/ycceh/(U54DK106857) to DSK. The funders did not play nay role in the study design, data collection and analysis, decision to publish, or preparation of the manuscript.

**Competing interests:** The authors have declared that no competing interests exist.

**Abbreviations :** BLTP, bridge-like lipid transfer protein; DEG, differentially expressed gene; DKO, double knockout; ER, endoplasmic reticulum; FACS, fluorescence-activated cell sorting; ISG, interferon stimulated gene.

of intracellular organelles, such as mitochondria and autophagic lysosomes, or to impaired autophagy, due to the absence of these lipid transport proteins. The surprising and striking synthetic effect resulting for the combined loss of VPS13A and VPS13C suggests that despite of the different localization of these two proteins, the lipid fluxes that they mediate are partially redundant.

## Introduction

A defining characteristic of eukaryotic cells is the presence of lipid-based intracellular membranes that delimit functionally distinct compartments. Growth of these membranes, as well maintenance of their distinct lipid composition and their property to undergo modifications in response to functional states require transport of lipids between them. This is achieved either via vesicular traffic or via protein-mediated transport [1]. A recently discovered mechanism of protein-mediated lipid transport involves rod-like proteins localized at sites of close appositions between intracellular membranes which allow bulk unidirectional transfer between them by a bridge like mechanism. These proteins, collectively called bridge-like lipid transfer proteins (BLTPs), include VPS13, the founding member of the family [2–6].

VPS13 family proteins are encoded by a single gene in yeast and by four genes in mammals which derived from successive gene duplications: *VPS13A, VPS13B, VPS13C,* and *VPS13D* [7,8]. These duplications have correlated with specialization of the four VPS13 proteins for distinct physiological roles, as demonstrated by their different localizations [9–20] and the different phenotypes resulting from their mutations [21–26]. The gene duplication that gave rise to *VPS13A* and *VPS13C* is the most recent in evolution as these two *VPS13* family members are the most similar to each other [7]. Their appearance correlated with the emergence of vertebrates as *Drosophila* and *C. elegans* only have a single gene closely related to both *VP13A* and *VPS13C* [7]. Mutations of such gene, called *Vps13* in flies and *vps-13A* in worms, lead to defective clearance of subsets of cell corpses in both worms (https://wormbase.org/species/c_elegans/gene/WBGene00011629#0-9fc-10) and flies [27] and also to neurodegeneration in flies [28]. In spite of their close similarity, mammalian VPS13A and VPS13C have different intracellular localizations. Based on current information, VPS13A is localized primarily at contacts between the endoplasmic reticulum (ER) and either mitochondria or plasma membrane (PM), but with differences depending on cell type and differentiation/functional state [10,12,17], while VPS13C is primarily localized at contacts between the ER and late endosomes/lysosomes [10,13,14], although other localizations have also been reported [16,24,29]. In humans, loss-of-function mutations of the two proteins are compatible with early normal life, but eventually are responsible for age-dependent neurodegenerative diseases. *VPS13A* mutations cause chorea-acanthocytosis (VPS13A disease), a Huntington's like condition due to degeneration of the caudate nucleus of the brain and abnormal red blood morphology [21,22,30]. In contrast, *VPS13C* mutations cause early onset Parkinson's disease, featuring rapid disease progression with

atrophy in multiple brain regions, including midbrain [24]. In mice, some defects are observed in either *Vps13a* KO [31–33] (https://www.mousephenotype.org/data/genes/MGI:2444304) or *Vps13c* KO animals (https://www.mousephenotype.org/data/genes/MGI:2444207), but no major neurodegenerative changes as in humans have been reported [31–34]. Given the neurodegeneration observed in flies that lack *Vps13*, i.e., the common ancestor of *VPS13A* and *VPS13C* [28], we considered that VPS13A and VPS13C may have partially redundant function, in spite of their predominantly different localizations. To explore such a possibility, we interbred *Vps13a* and *Vps13c* KO mice to generate double KO (DKO) mice.

Surprisingly, this intercrossing led to embryonic lethality with arrest of development at midgestation. A major defect was an impairment of hematopoiesis. Analysis of DKO erythroid precursor cells isolated from E12.5 embryonic liver revealed downregulation of hematopoietic genes and upregulation of interferon stimulated genes (ISGs). Further analysis of both fetal liver tissue of DKO embryos and of partially differentiated DKO human erythroblast-like cells demonstrated activation of the dsRNA sensing innate immunity RIG-I pathway. We speculate that activation of innate immunity may result from loss of the integrity of the membranes of intracellular organelles or to impaired autophagy, and that erythroid precursor cells are especially sensitive to loss of VPS13A and VPS13C.

## Results

### Embryonic lethality of *Vps13a*/*Vps13c* double knockout mice

To generate a *Vps13a* and *Vps13c* double KO (DKO) mice line, we first established in the lab *Vps13a* KO mice starting from cryopreserved sperm of *Vps13a*tm1a obtained from the European Mutant Mouse Archive (S1A Fig; see "Materials and methods"). *Vps13a* KO mice were born with mendelian distribution and did not exhibit any obvious phenotypic defects except male infertility [31] and immunoblot analysis of cortical brain tissue from these mice confirmed the absence of VPS13A, as previously reported for the same mouse line [32] (S1B Fig).

We next interbred these mice with previously described *Vps13c* KO mice also generated with sperms obtained from the European Mutant Mouse Archive [34]. Surprisingly, these crossings did not result in any DKO offsprings, indicating embryonic lethality of this genotype (S1C Fig). Subsequent analysis of timed pregnancies of *Vps13a*+/−/*Vps13c*−/− mice intercrossing revealed mendelian distribution of all three expected genotypes at embryonic day 12.5 (E12.5), but not at E15, when DKO embryos were not found. Moreover, DKO embryos at E12.5 were smaller and paler than mice of the other two genotypes (*Vps13a*+/+/*Vps13c*−/− and *Vps13a*+/−/*Vps13c*−/−). Mice with these two genotypes were similar to WT embryos at the same developmental stage and did not exhibit obvious differences from each other (S1C Fig, Fig 1A and 1B). We conclude that most likely DKO fetuses get reabsorbed after E12.5 and that VPS13A and VPS13C have a redundant function in some essential process(es) during embryonic development.

### Hematopoiesis defects in DKO mouse embryos

Compared to control animals, DKO fetuses were smaller and paler, with delayed development of ears and digits, poorly defined vasculature and a smaller, paler liver (Fig 1A–D). These findings suggested a defect of either hematopoiesis or of vasculature development or of both. Hematopoiesis is the process in which stem cells proliferate and differentiate to produce mature red blood cells. Hematopoietic progenitors arise developmentally from vascular endothelium, initially expressing the endothelial cell marker cluster of differentiation 31 (CD31) [35]. Thereafter, multilineage hematopoiesis occurs in sequential locations during development, with fetal liver becoming the key hematopoietic site at E11–12 in mice [36]. Moreover, immunostaining of sections of E12.5 DKO and littermate embryos for CD34, a marker for both hematopoietic progenitor cells and vascular endothelial progenitor cells [37], revealed little CD34 immunoreactivity in the DKO liver (Fig 1E–1H). In contrast, immunostaining for the vascular endothelial cell marker CD31 [38] did not reveal obvious differences between DKO and littermate embryos, suggesting that hematopoiesis was the primary defect in DKO embryos (Fig 1I–1L). We also performed immunostaining for cleaved caspase-3 [39] to assess the presence of apoptotic cell death at E12.5, either due to cell autonomous effects of VPS13A and VPS13C deficiency or to ischemia resulting from defects

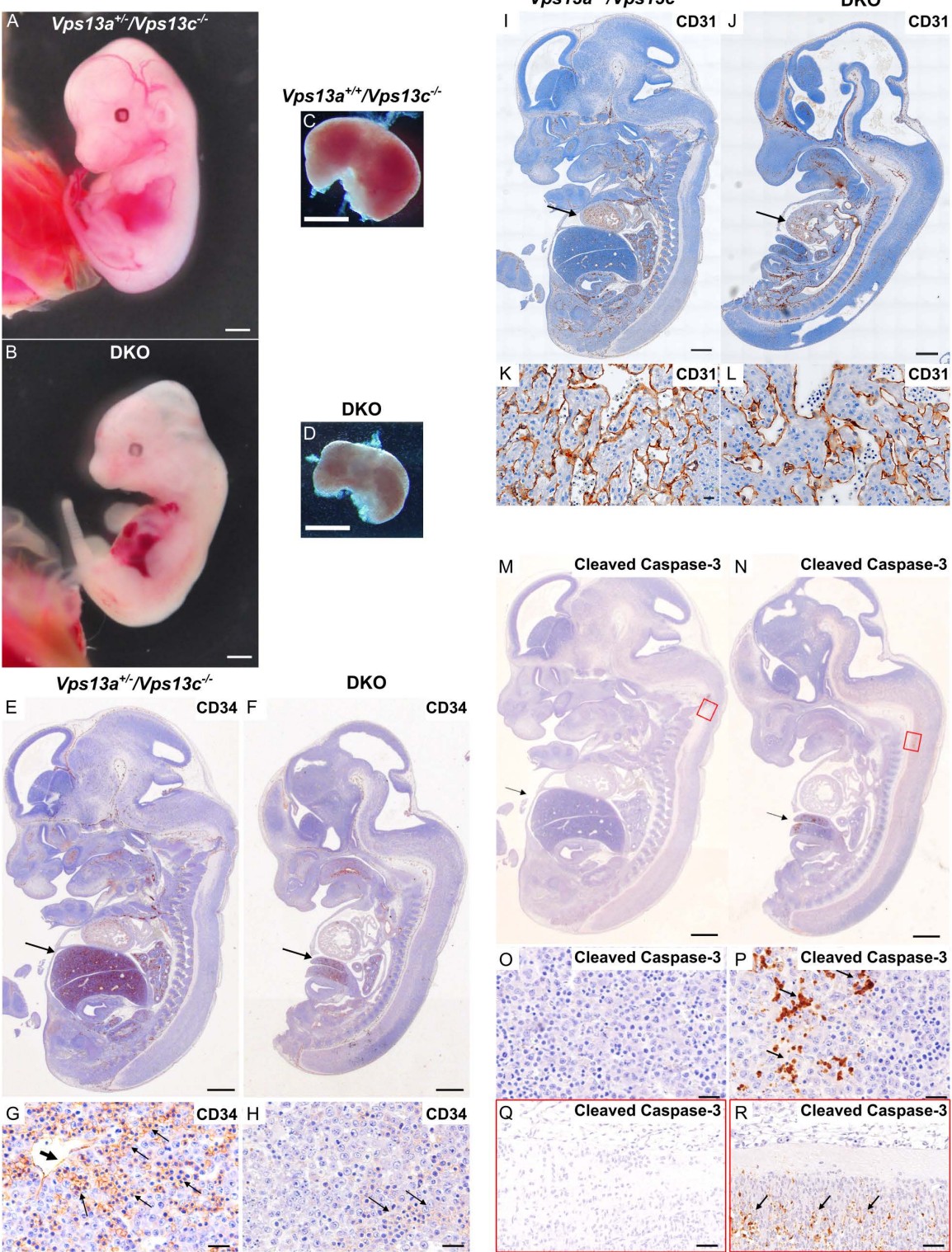

**Fig 1. Gross fetal pathology and histology of E12.5 DKO embryos and controls.** All controls (see genotypes on the figure) are KO for *Vps13c* and WT or heterozygous for *Vps13*a KO. **(A, B)** Comparison of a DKO embryo to a *Vps13a+/-/Vps13c-/-* embryo. The DKO embryo is smaller and paler, with delayed development of ears and digits, poorly visible vasculature and a smaller, paler liver. **(C, D)** Livers dissected from DKO and *Vps13a+/+/Vps13c-/-* embryos. **(E-H)** H&E stained longitudinal section of DKO and control fetuses processed for anti-CD34 immunohistochemistry. The DKO liver is much

smaller [arrows in **(E)** and **(F)**] with reduced CD34 immunoreactivity **(G, H)** where the CD34 immunoreactivity on vascular endothelia (thick arrow, **G**) and hematopoietic precursors (thin arrows, **G and H**) is shown in brown. **(I–L)** H&E stained longitudinal section of DKO and control fetuses processed for anti-CD31 immunohistochemistry showing no difference in CD31 immunoreactivity in cardiac tissue **(arrows, I, J)**. Sections of the heart are shown at higher magnification in panels **K and L**. **(M–R)** H&E stained longitudinal sections of DKO and control fetuses processed for anti-cleaved caspase-3 immunohistochemistry revealing presence of apoptotic cells, primarily in liver (arrow) and in the lower portion of the brain stem (areas surrounded by red rectangles) in panels **(M)** and **(N)**. These regions are shown at higher magnification in panels **O and P** and in panels **Q and R**, respectively. Cleaved caspase-3 immunoreactivity is only present in DKO tissues. Scale bar: 1 mm **(A, B, C, and D)**; 500 μm **(E, F, I, J, M, and N)**; 20 μm **(G, H, K, L, O, P, Q, and R)**. Raw images of this figure can be found at https://doi.org/10.5281/zenodo.15375803.

in erythropoiesis. We found that such immunoreactivity was nearly absent in control (*Vps13a*$^{+/-}$/*Vps13c*$^{-/-}$) embryos, but clearly detectable in DKO embryos (Fig 1M and 1N), especially in fetal liver and the lower portion of the brain stem (Fig 1O–1R) consistent with the subsequent arrest of embryonic development.

### Defect in erythrocyte differentiation and activation of innate immunity in DKO erythroid progenitor cells

To gain further insight into the hematopoietic defect, we profiled erythroid progenitors by fluorescence-activated cell sorting (FACS) in E12.5 livers of WT and mutant embryos derived from interbreeding of *Vps13a*$^{+/-}$/*Vps13c*$^{-/-}$ mice. Fetal livers were harvested, and dissociated cells were stained for FACS analysis (Fig 2A). A lower percentage of live cells was obtained from DKO livers than from livers of *Vps13a*$^{+/+}$/*Vps13c*$^{-/-}$ and *Vps13a*$^{+/-}$/*Vps13c*$^{-/-}$ littermates, which in turn was similar to the percentage of live cells obtained from the livers of WT embryo at the same developmental stage (Figs 2B and S2A). DAPI negative live cells were then assessed with a cocktail of lineage-specific markers labeling non-erythroid cells to negatively select such cells and focus on distinct stages (S0 to S5) of erythroid maturation based on differential expression of CD71 (transferrin receptor, high in erythroid precursors) and Ter119 (a surface protein high in mature erythrocytes) [40,41] (Fig 2C). Cells harvested from WT fetal liver comprised erythroid precursors and immature erythroblasts (S0 to S4) with S3 cells being the most abundant, as expected [40] (Fig 2C top panel and 2D). Similar results were obtained using livers from *Vps13a*$^{+/+}$/*Vps13c*$^{-/-}$ and *Vps13a*$^{+/-}$/*Vps13c*$^{-/-}$ littermates of DKO embryos (S2B Fig). However, most the DKO cells were at the S0 stage (i.e., CD71 and Ter119 negative) indicating impaired erythroid development and maturation (Fig 2C bottom panel, and 2D). We conclude that precursors of erythroid cells are especially sensitive to the combined absence VPS13A and VPS13C.

To further elucidate molecular mechanisms underlying impaired erythroid maturation in DKO embryos, we isolated S0 cells from E12.5 WT and DKO fetal livers following an established protocol [40,42] and compared their transcriptomic profiles by RNA sequencing (RNAseq) (Fig 3A). Consistent with this previous study [42] showing that the majority of S0 cells are erythroid precursors, we found that the transcriptomic profiles of our WT and DKO S0 cells are most similar to the transcriptomic profile of erythroblasts (S2C Fig). Specifically, in both WT and DKO S0 cells, we observed high expression level of *Klf1*, *Gata1*, *Gata2*, *Ldb1*, and *Lmo2*, key transcriptional regulators driving erythroid fate commitment and erythroid maturation (S2D Fig) [43–51]. Hemoglobin genes were also highly expressed (S2D Fig) with higher expression of embryonic β-globins in DKO (i.e., *Hbb-bh1*, *Hba-x*, *Hbb-y*), further implying that loss of VPS13A and VPS13C affects definitive erythropoiesis.

Among the top 20 significantly differentially expressed genes (DEGs) between WT and DKO S0 cells were 7 members of the interferon stimulated gene (ISG) family [52], implying activation of innate immunity in DKO S0 cells (Fig 3B). Accordingly, pathway analysis of these genes showed that upregulated genes were enriched in "interferon gamma" and "interferon alpha" responses [MSigDB database [53]], while downregulated genes were enriched in "hematopoietic cell lineage" pathways [KEGG database [54]], in agreement with the defective hematopoiesis phenotype of DKO embryos (Figs 3C, 3D, and S2E). Significantly downregulated genes in DKO cells included *Kit*, an essential gene for erythropoiesis [55] and *Cd34*, which is consistent with the reduced levels of CD34 immunoreactivity in the fetal liver of DKO reported above.

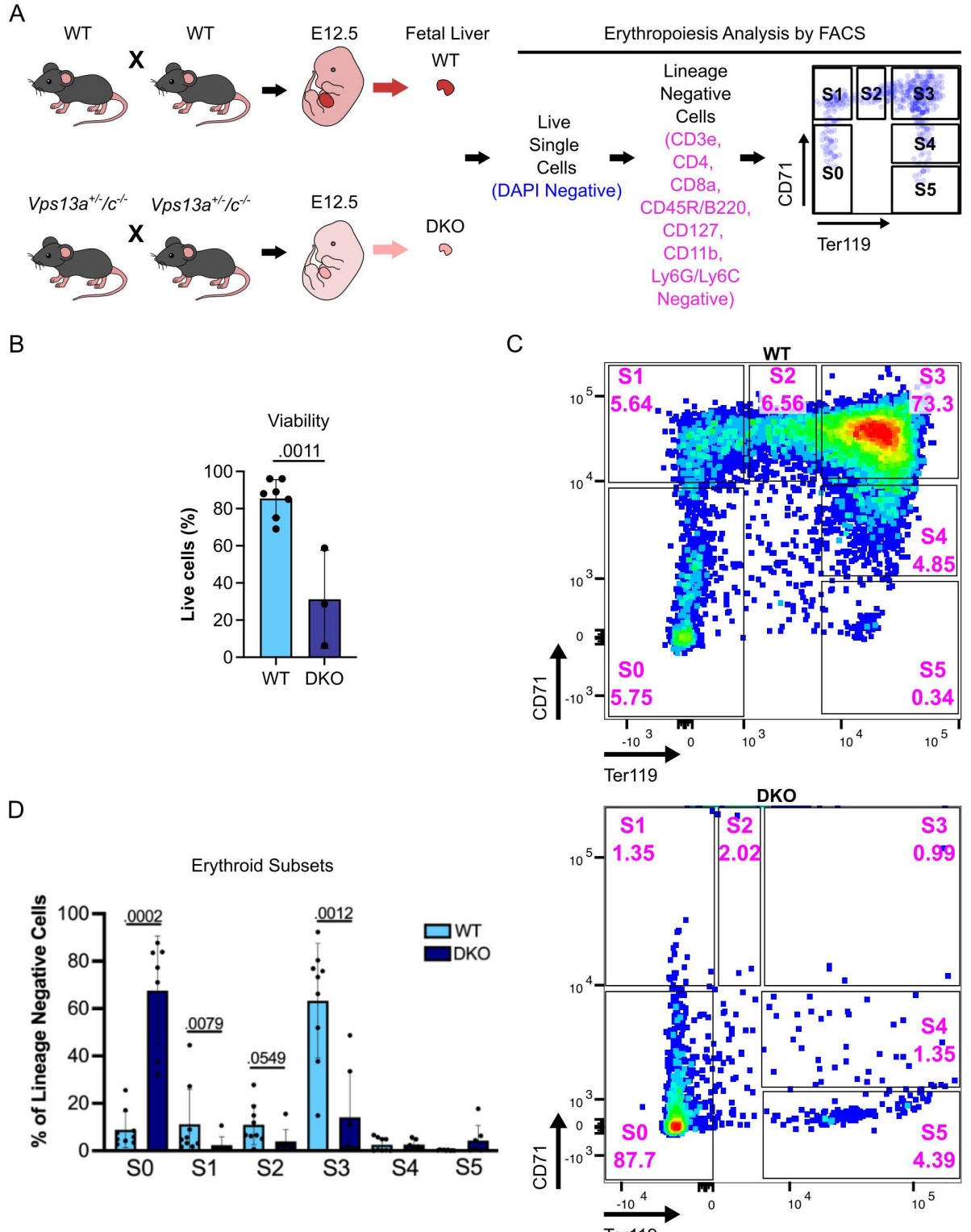

**Fig 2. The DKO of *Vps13a and Vps13c* abrogates embryonic erythroid maturation. (A)** Experimental design for the analysis of erythropoiesis via FACS. **(B)** Viability of cells derived from WT and DKO fetal livers assessed by trypan blue staining. Each dot represents one fetal liver with the specific genotype. **(C)** Representative FACS plots of WT (top panel) and DKO (bottom panel) E12.5 fetal livers, respectively, showing subsets of erythroid cells

with S0 through S5 representing progressive erythroid maturation stages. **(D)** Percentage of each erythroid subset within the lineage negative gate. Each dot represents one fetal liver ($n = 9$ for WT and $n = 7$ for DKO). Results are presented as average ± standard deviation (SD) of each subtype of erythroid cells with *p* values indicated for statistically significant differences between WT and DKO embryos. Raw data of this figure can be found at https://doi.org/10.5281/zenodo.15375803.

## Upregulation of the RIG-I and MDA5 dependent RNA sensing pathway in DKO cells

Enhanced transcription of ISGs suggests activation of cytosolic nucleic acid sensors [52,56]. Further inspection of the list of upregulated transcripts in DKO S0 cells revealed approximately 3-fold upregulation of the cytosolic double stranded RNA (dsRNA) sensors MDA5 and RIG-I (encoded by the *Ifih1* and *Ddx58* genes, respectively) suggesting cytosolic leakage of mitochondrial dsRNA [57–61] (Fig 4A). In agreement with RNAseq data, immunoblot analysis of control and DKO E12.5 fetal livers revealed increased levels of MDA5 and RIG-I relative to WT. Moreover, we examined protein levels of the well-established ISG, *Ifitm3* [62], which is the product of one of the top upregulated transcripts in DKO S0 cells, and we observed its upregulation in DKO liver (Fig 4B).

To further assess the impact of the combined absence of VPS13A and VPS3C on innate immunity in a human cell line of erythroblastic lineage, we utilized K562 cells which differentiate into pro-erythroblast-like cells upon treatment with hemin [12]. To this aim we knocked out both *VPS13A* and *VPS13C* in K562 cells. Consistent with mouse fetal liver data, we observed that MDA5 and IFITM3 proteins are highly upregulated both in naïf DKO K562 cells and after hemin-induced differentiation (Fig 4C–4E), while the cytosolic DNA sensing cGAS-STING pathway was not activated relative to controls (WT K562 cells), as determined by phospho-STING levels (Fig 4E).

## Discussion

We report here that while absence of either VPS13A or VPS13C is compatible with nearly normal life in mice, and with lack of obvious pathology in young humans, the combined absence of these two proteins in mice results in arrest of embryonic development at around 12.5 embryonic day. Clearly cellular life, and even the life of an early mouse embryo, can occur without both proteins, as DKO mouse fetuses develop up to midgestation. Moreover, flies and worms that lack the single common ancestor gene of *VPS13A* and *VPS13C* have only selected defects, including impaired elimination of subsets of dead cells in flies [27] and worms (https://wormbase.org/species/c_elegans/gene/WBGene00011629#0-9fc-10) and neurodegeneration in flies [28]. These results imply that VPS13A and VPS13C have essential but redundant function in a process, or processes, whose failure(s) result(s) in the arrest of mammalian organism development. Moreover, our results suggest that failure of erythropoiesis, leading to impaired oxygen supply to tissues, plays a critical role in such arrest, although the role of other factors cannot be excluded. While a major phenotype observed the DKO embryos is a small liver, such defect is likely explained by defective erythropoiesis, as cells of the erythroblastic lineage are by far the most important contributor to the liver mass at this embryonic stage [63]. This phenotype and resulting embryonic death have been repeatedly observed in mice deficient in genes required for embryonic erythropoiesis [64,65]. Moreover, although in the characterization of the erythropoiesis defect we focused on definitive erythropoiesis in E12.5 DKO embryos, the paler color of these embryos indicates a severe defect also in primitive erythropoiesis, which generates circulating primitive erythroid cells before E12.5. It is therefore likely that lack of both *Vps13a*/*Vps13c* causes the impairment of both primitive and definitive erythropoiesis.

We also found evidence for the activation of innate immunity in erythrocyte precursors that fail to differentiate, as RNA-seq identified several ISGs among the top upregulated genes in these cells. Whether such activation plays a role in the impairment of erythrocyte differentiation, or whether the two events are independent consequence of the combined loss of VPS13A and VPS13C, remains to be determined. VPS13A and VPS13C are thought to participate in the delivery of

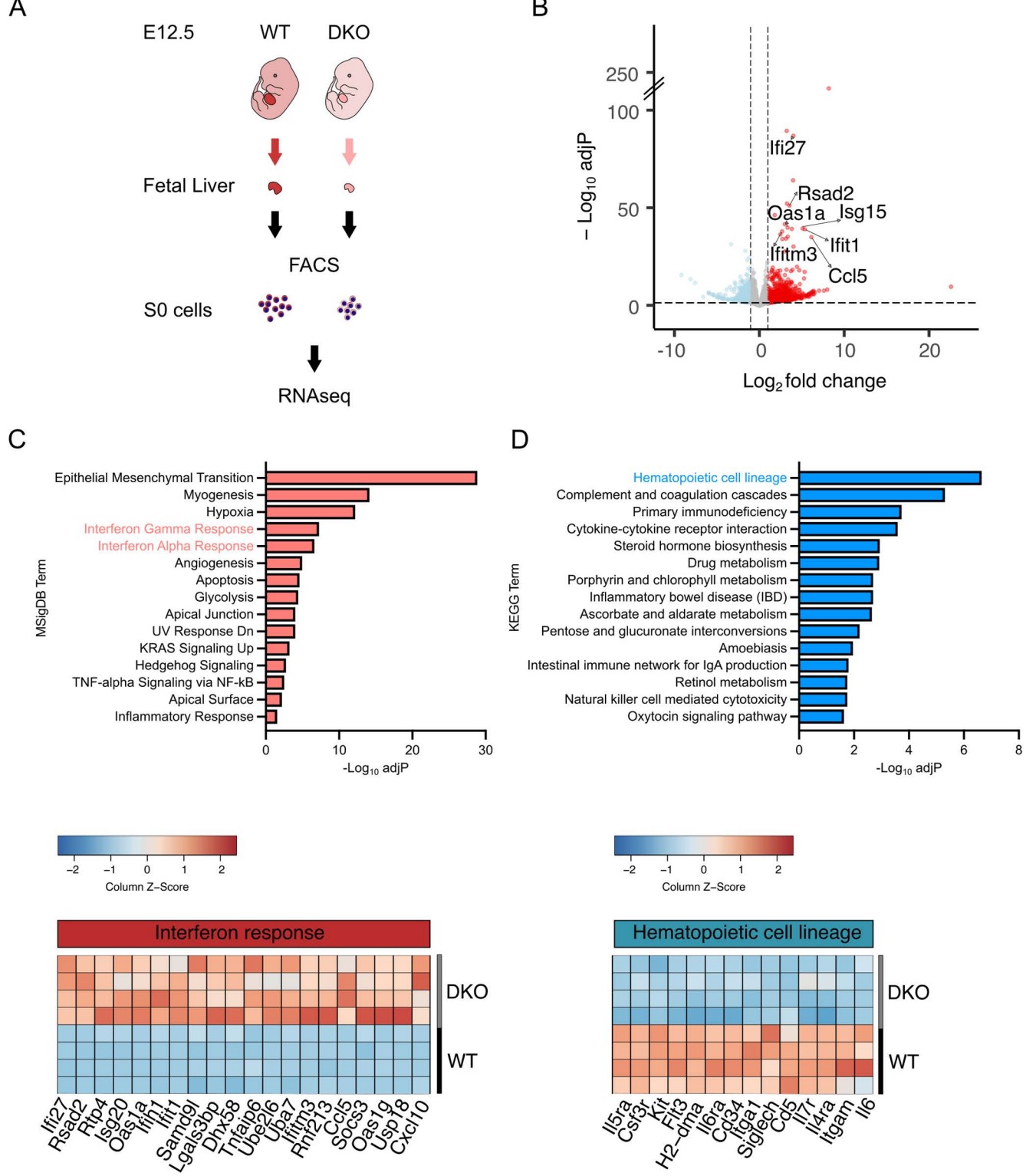

**Fig 3. Transcriptomic profile of DKO erythroid precursor S0 cells from E12.5 fetal liver compared to WT. (A)** Experimental design. **(B)** Volcano plot comparing Differentially Expressed Genes (DEGs) in DKO S0 cells relative to WT. Blue and red dots represent significantly downregulated or upregulated genes, respectively, in DKO [log₂fold change < −1 or > 1 and adjusted *p*-values (adj *P*) < 0.05]. ISGs in the top significantly 20 upregulated genes

are indicated. **(C)** Bar plot showing the top 15 significantly upregulated pathways (according to adjusted *p*-values) in S0 cells according to the MSigDB database (top) and heatmap showing the top 20 significantly upregulated genes in the "interferon (gamma/alpha) response" pathways (bottom). **(D)** Bar plot showing the top 15 significantly downregulated pathways in S0 cells according to the KEGG database (top) and heatmap showing the genes of the "hematopoietic cell lineage" pathway (bottom). Raw data of this figure can be found at https://doi.org/10.5281/zenodo.15375803.

lipids to mitochondria [10] and to lysosomes [14,34], respectively. Thus, a potential explanation for the activation of innate immunity in erythrocyte precursors is the impairment of the integrity of the membranes of mitochondria and autophagic lysosomes, with the release of mitochondrial nucleic acids into the cytosol. The importance of VPS13 for the integrity of mitochondrial membranes to prevent escape of nucleic acids was demonstrated in yeast, where the single VPS13 protein in this organism was one of the hits in a screen for proteins required to block DNA escape form these organelles [66]. Similarly, presence of mitochondrial DNA in the cytosol and activation of the DNA sensing cGAS-STING pathway of innate immunity was detected in *VPS13C* KO HeLa cells [34]. In these cells nucleic acid may leak out of lysosomes digesting mitochondria, although an indirect effect of lysosome defects on mitochondria integrity could not be excluded. In the present study we found evidence for activation of the dsRNA sensing pathway involving RIG-I and MDA5 in the absence of VPS13A and VPS13C, rather than of the cGAS-STING pathway. The mRNAs encoding these two proteins were elevated in S0 erythrocyte precursors in the fetal liver and their protein levels were increased in homogenates of fetal liver. We confirmed elevated protein levels of RIG-I and MDA5 in K562 *VPS13A*/*VPS13C* double KO cells, i.e., cells which can be differentiated in proerythroblasts by hemin. In spite of this difference in the predominant nucleic acid sensing pathway activated by loss of VPS13 function in different model systems, our findings are consistent with a role of VPS13 family proteins in preserving membrane integrity.

It is also possible that a primary defect in autophagy may be responsible for, or contribute to, the activation of innate immunity in erythrocytes precursors lacking both VPS13A and VPS13C, as roles of VPS13 family proteins in autophagy have been reported [33,67–70]. For example, the single yeast Vps13 protein cooperates with Atg2 (a bridge-like lipid transfer protein of the VPS13 protein family) in the growth of the autophagic membrane [67–69] and both mammalian VPS13A and VPS13C were proposed to participate in this process via an interaction with the scramblase ATG9, a well-known interaction partner of ATG2 [71]. Moreover, VPS13A was implicated in autophagy by studies of human and mouse skeletal muscle [33]. Erythrocyte precursors may be especially sensitive to a defect in autophagy, as their maturation involves the autophagic clearing of intracellular organelles, including mitochondria [72–74]. A defect in this process would prevent appropriate sequestration of mitochondrial nucleic acids and other toxic material resulting in activation of innate immunity [75]. Accordingly, in an unbiased proteomics screen for elevated proteins in autophagy deficient cells, the RIG-I signaling pathway was one of the top hits [76] and the RIG-I pathway was enhanced in cells deficient in the ATG5-ATG12 complex, which is required for autophagy [77,78].

The striking synthetic effect of the loss of VPS13A and VPS13C in mice development is surprising, given that most available evidence suggests predominantly distinct localizations of VPS13A and VPS13C [4,10,12,16,79], and that *Vps13a* and *Vps13c* single KO mice have nearly normal life. Even in human lack of either protein alone does not affect early life, but result in age-dependent neurodegenerative diseases. However, major redundancies and intersections in lipid fluxes mediated by lipid transport proteins have been reported [80], so that a defect in lipid transport at one intracellular site can be bypassed by transport at other sites. Thus, VPS13A and VPS13C may have partially redundant functions even if they are differentially localized. It also remains possible that VPS13A and VPS13C have yet uncharacterized overlapping subcellular localizations (for example a localization of VPS13C at ER-mitochondria contacts, where VPS13A is also present, had been suggested [10,24]). Moreover, both proteins have been found at ER-lipid droplets contacts [10,16], although the functional significance of this localization remains unclear. So far, localization studies of these two proteins have relied primarily on their overexpression (tagged proteins) in cell lines, given the limitations of currently available antibodies. Future studies of endogenously tagged proteins and in different tissues will be needed to obtain a full picture

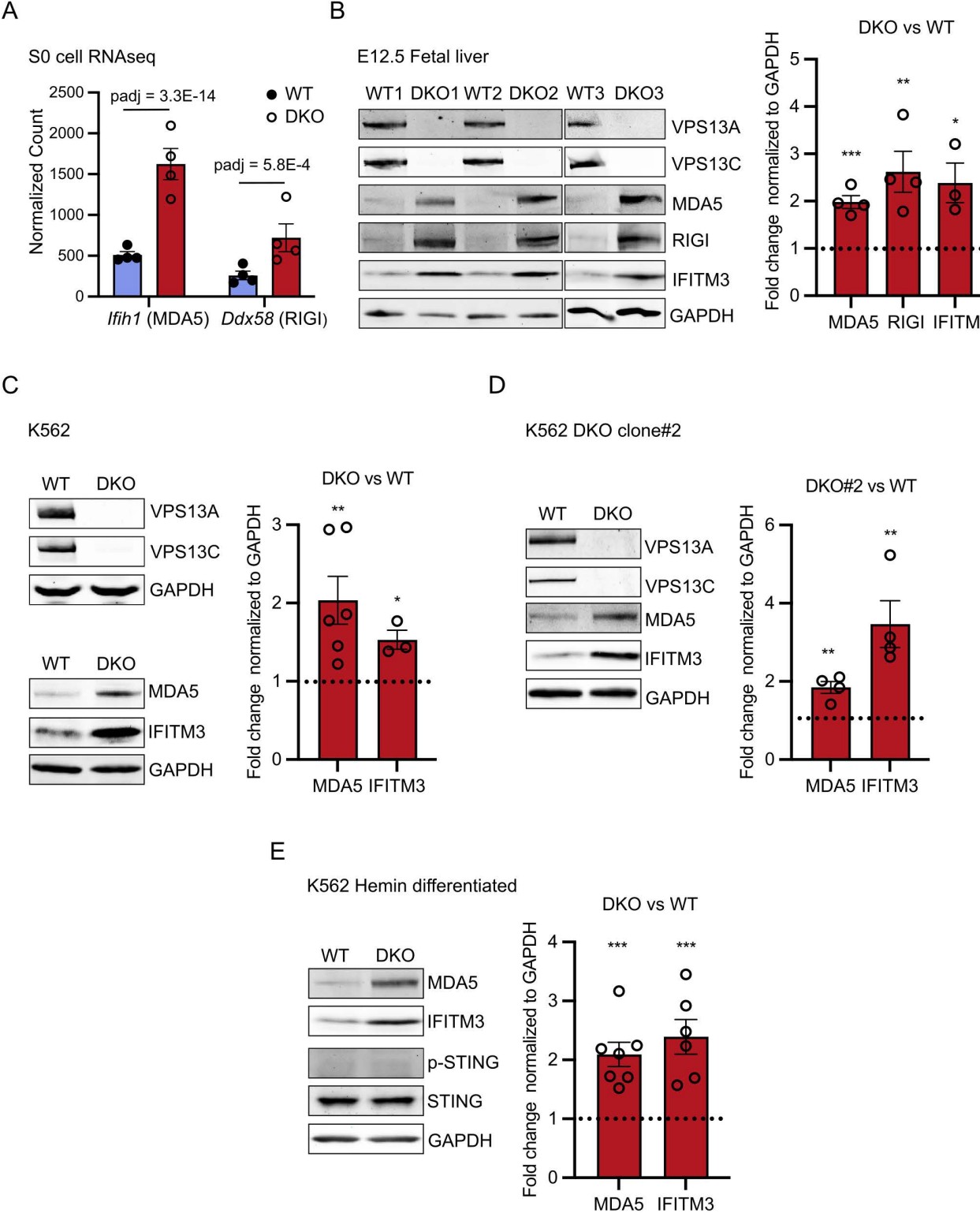

**Fig 4. Activation of the dsRNA sensing pathway in DKO fetal liver and DKO K562 cells. (A)** Bar plot quantifying transcript levels of the dsRNA sensing proteins MAD5 and RIGI from S0 cells RNAseq data in WT and DKO cells. **(B)** Immunoblots for the indicated proteins of whole fetal liver lysates from E12.5 WT and DKO embryos (left). IFITM3 is a representative ISG. GAPDH was used as a control. Protein levels based on the immunoblots were

quantified as fold change in DKO relative to WT (right). **(C)** Immunoblots for the indicated proteins of WT and DKO K562 cells (left). Protein levels were quantified as fold change in DKO relative to WT (right). **(D)** Immunoblots of lysates from WT and DKO K562 clone #2 with probed antibodies indicated (left). Protein levels were quantified as fold change comparing DKO#2 to WT (right). **(E)** Immunoblots of lysates from hemin differentiated WT and DKO K562 cells with probed antibodies indicated (left). Protein levels were quantified as fold change comparing DKO to WT (right). Protein levels were quantified as fold change comparing DKO to WT (right). Data represent mean ± SEM; unpaired $t$ test; * $P < 0.05$; ** $P < 0.01$; *** $P < 0.001$. Raw data of this figure can be found at https://doi.org/10.5281/zenodo.15375803.

of the cellular and subcellular distributions of these proteins under control conditions and in different functional states towards a better understanding of their roles in physiology and disease.

## Materials and methods

### Mouse

All animal studies were conducted in compliance with guidelines from the US Department of Health and Human Services Guide for the Care and Use of Laboratory Animals under Yale Institutional Animal Care and Use Committee protocol #2024-07422. Mice were maintained in a C57BL/6J genetic background and housed in rooms on a 12-h dark/light cycle interval with food and water available ad libitum. $Vps13c^{-/-}$ mice (MGI:7489965) were previously generated in our lab [34]. Cryopreserved sperm from $Vps13a^{tm1a}$ mice was obtained from the European Mutant Mouse Archive (http://www.informatics.jax.org/allele/MGI:4842502) and sequentially bred with FLP recombinase and β-actin-Cre transgenic mice to generate the $Vps13a^{tm1d}$ ($Vps13a^{-/-}$) allele (RRID: MGI:8190267). Loss of function of $Vps13a$ was confirmed both at the genome level by Sanger sequencing using Transnetyx and at the protein level by immunoblotting. For biochemical studies, mice were euthanized in a $CO_2$ chamber, embryos and fetal livers were immediately dissected on ice before being processed as described in the immunoblotting section.

### Pathological analysis of embryos

Pregnant dams were euthanized at estimated embryonic day 12.5 using a carbon monoxide chamber (filled gradually to 70%) followed by creation of pneumothorax. The uterus was extracted in its entirety, after which fetuses ($n = 6$ per litter) and their membranes were dissected intact from the uterus. The tail of each fetus was harvested for genotyping. Fetuses were fixed in Bouin's solution for 48 h, followed by bisection along the sagittal plane. Cassetted tissues were submitted for standard paraffin embedding and processing, followed by generation of 20–30 sections (5 μm thick) collected at 20 μm intervals. Sections were stained with hematoxylin and eosin. Immunoperoxidase stains were performed using unstained 5 μm paraffin sections at the Yale University Department of Pathology using primary antibodies at a 1:100 dilution. After deparaffinization and rehydration, antigen retrieval was performed using sodium citrate buffer (10 mM Sodium Citrate, 0.05% Tween 20, pH 6.0) at 95–100°C for 10 min. Immunostaining was performed using a Dako autostainer. Label was visualized by 0.05% 3′,3′-diaminobenzidine (DAB) as a chromogen, precipitated by 0.01% hydrogen peroxide. Light microscopic images were taken using a Zeiss Axioskop and with Axiocam MrC camera. A protocol is also available at https://www.protocols.io/view/pathological-analysis-of-mosue-embryos-kxygx4emdl8j/v1.

### Fetal liver flow cytometry analysis and FACS

Male and female heterozygous mice were set up for mating in the evening. Males were removed from the cage on the following day, which was considered E0.5 for the embryos of pregnant females. Pregnancy was assessed by monitoring the female weight. At E12.5, embryos were removed, and tail clips used for genotyping. Fetal livers were isolated using a dissecting microscope to assure no contamination by surrounding tissues. For FACS analysis, fetal livers were dissociated into single cell suspensions and passed through a 100 μm nylon strainer to eliminate cell clumps and debris. Percentages of live cells were determined by trypan blue staining. Cells were then counted and stained with a lineage

cocktail (biotinylated antibodies against CD3e, CD4, CD8a, CD45R/B220, CD127, CD11b, Ly6G/Ly6C) followed by streptavidin-PE-Cy7 (Biolegend, Cat# 405206), anti-CD71-FITC, anti-Ter119-PE, and DAPI (BD, Cat# 564907). The gate strategy was conducted according to [40]. Samples were analyzed on LSRII and BD FACSAriaII analyzers for flow cytometric analysis and FACS sorting, respectively. A protocol is also available at https://www.protocols.io/view/fetal-liver-flow-cytometry-analysis-and-facs-q26g7nbrklwz/v1.

## Antibodies

Primary antibodies used: CD34 (Thermo Fisher Scientific Cat# MA1-22646, RRID:AB_558179), CD31 (Cell Signaling Technology Cat# 77699, RRID:AB_2722705), Caspase-3 (R and D Systems Cat# MAB835, RRID:AB_2243951), CD3e (BD Biosciences Cat# 553060, RRID:AB_394593), CD4 (BD Biosciences Cat# 553649, RRID:AB_394969), CD8a (BD Biosciences Cat# 553028, RRID:AB_394566), CD45R/B220 (BD Biosciences Cat# 553086, RRID:AB_394616),CD127 (BD Biosciences Cat# 555288, RRID:AB_395706), CD11b (BD Biosciences Cat# 553309, RRID:AB_394773), Ly6G/Ly6C (BD Biosciences Cat# 553125, RRID:AB_394641), CD71-FITC (BioLegend Cat# 113806, RRID:AB_313567), Ter119-PE (BD Biosciences Cat# 553673, RRID:AB_394986), VPS13A (Proteintech Cat# 28618-1-AP, RRID:AB_2918183), VPS13C (custom [34], Proteintech, RRID: AB_3711218), MDA5 (Cell Signaling Technology Cat# 5321, RRID:AB_10694490), RIGI (Proteintech Cat# 25068-1-AP, RRID:AB_2879881), IFITM3 (Cell Signaling Technology Cat# 59212, RRID:AB_2799561), GAPDH (Proteintech Cat# 60004-1-Ig, RRID:AB_2107436), STING (Cell Signaling Technology; Cat# 80231, RRID:AB_2799947), Phospho-STING (S366; Cell Signaling Technology; Cat# 19781, RRID:AB_2737062).

Secondary antibodies used: IRDye 800CW Donkey anti-Rabbit IgG (LI-COR Biosciences Cat# 926-32213, RRID:AB_621848) and IRDye 680LT Goat anti-Mouse IgG (LI-COR Biosciences Cat# 926-68020, RRID:AB_10706161).

## DNA plasmids and oligos

For CRISPR mediated gene knockout in K562 cells, candidate guide RNAs (gRNAs) against the human *VPS13A* genomic locus were: (5′-CCTTCAAAATCTGAGCATGC-3′ for PX459 cloning) and (5′-ACTATAGCCAATTGCTTCATAGA-3′ for PY30 cloning). gRNAs were ordered as complementary single-stranded oligonucleotides from Integrated DNA Technologies (IDT) then cloned into the PX459 Cas9 plasmid (Addgene plasmid #62988) or PY30 Cas12a plasmid (Addgene plasmid #84745) using a one-step ligation protocol [81], and gRNAs were sequence verified using the U6 forward promoter. A PX459 plasmid (Addgene plasmid #62988) targeting *VPS13C* was generated previously. All oligos were purchased from IDT.

## Cell culture and transfection

K562 cells (gift of Patrick Gallagher, Yale University) were cultured at 37°C in 5% $CO_2$ and RPMI 1640 containing 10% FBS, and 2 mM GlutaMAX (all from Gibco). Transfection of plasmids was accomplished using FuGene 4K Transfection Reagent (Promega). For a detailed protocol for cell culture, and transfection, see https://doi.org/10.17504/protocols.io.e6nvwdk4dlmk/v1.

## Generation of *VPS13A* and *VPS13C* DKO K562 cell lines

K562 cells were transfected with plasmids (either PX459 Cas9 system for DKO clone #1 or PY30 Cas12a system for clone #2) containing guide RNAs targeting *VPS13A* using FuGene 4K. Cells were selected in complete culture media containing 3 µg/ml puromycin 24 h after transfection and the medium was replaced with fresh puromycin-containing medium 48 and 72 h after transfection. After three days of puromycin selection, single clones were obtained using serial dilution and then screened by immunoblotting to identify VPS13A negative cells. Two *VPS13A* KO clones were transfected with PX549 containing a guide RNA targeting *VPS13C* using FuGene 4K. Cells were then selected in complete culture

media containing 3 µg/ml puromycin 24 h after transfection and the medium was replaced with fresh puromycin-containing medium 48 and 72 h after transfection. After three days of puromycin selection, single clones were obtained using serial dilution and then screened by western blotting for the absence of both VPS13A and VPS13C. A detailed method for the generation of KO cells is available at https://doi.org/10.17504/protocols.io.eq2lynx5wvx9/v1.

## Immunoblotting

Cultured cells were lysed through repeated pipetting in 2% SDS. Mouse tissue samples were immersed in 2% SDS and homogenized in a glass homogenizers followed by repeat pipetting. Cell lysates were further sonicated for 30 seconds to break DNA and eliminate viscosity of the samples. Total protein content was then measured by the Pierce BCA assay (Thermo Fisher Scientific). Equal amount of proteins were mixed with SDS loading buffer [final: Bromophenol blue (0.05%), 15% β-mercaptoethanol, Glycerol (10%), SDS (sodium dodecyl sulfate; 2%), Tris-Cl (0.05 M, pH 6.8)] and denatured at 70°C for 3 min. Samples were separated on 4–20% tris-glycine mini gels (Thermo Fisher Scientific) before transfer to nitrocellulose membranes at 4°C for one hour at 100 volts in transfer buffer containing 25 mM Tris, 192 mM glycine, 20% methanol in milliQ water. Transferred membranes were blocked in 5% BSA in Tris-buffered Saline (TBS) containing 0.1% Tween-20 (TBST) for 1 h. Membranes were then incubated with primary antibodies in 5% BSA in TBST overnight at 4°C. The next day, membranes were washed 3 times in TBST and then incubated with secondary antibodies conjugated to IRdye 800CW or IRdye 680LT (Licor, 1:10,000) in 5% BSA in TBST at RT for 1 h, washed 3 times in TBST, and then imaged using a Licor Odyssey Infrared Imager. A detailed protocol can be accessed on protocols.io at https://doi.org/10.17504/protocols.io.bp2l6be9zgqe/v1.

## RNAseq

Total RNA was purified using the RNAqueous-Micro Total RNA Isolation Kit (Thermo Fisher Scientific). RNA quality was determined by estimating the A260/A280 and A260/A230 ratios by nanodrop. RNA integrity was determined by running an Agilent Bioanalyzer gel, which measures the ratio of the ribosomal peaks. Samples with RIN values of 7 or greater were used for library preparation. Using the NEBNext Single Cell/Low Input RNA Library Prep Kit from Illumina (E6420L), a normalized RNA input between 200 pg and 50 ng was used to generate cDNAs with the template switching method. The NEB Ultra II FS workflow was used for fragmentation, end repair, dA-tailing, adapter ligation and PCR. Indexed libraries that met appropriate cut-offs for both quantity and quality were quantified by qRT-PCR using a commercially available kit (KAPA Biosystems) and insert size distribution was determined with the LabChip GX or Agilent Bioanalyzer. Samples with a yield of ≥0.5 ng/µl were used for sequencing. The libraries underwent 150-bp paired-end sequencing using an Illumina Novaseq X at the Yale Center for Genome Analysis (YCGA), generating an average of 82 million paired-end reads per library. Adapter sequences, empty reads, and low-quality sequences were removed. Reads were trimmed using fastp v0.21.0 [82] using default parameters. Trimmed reads were mapped to the mouse reference genome (mm10) with HISAT2 v2.2.1 [83]. Alignments with a quality score below 20 were excluded from further analysis. Gene counts were produced with StringTie v1.3.3b [84] and the Python script "prepDE.py" provided in the package (Python version 3.11.8). StringTie was limited to reads matching the reference annotation GENCODE V15 [85]. After obtaining the matrix of read counts, differential expression analysis was conducted, and normalized counts were produced using DESeq2 [86]. p-values were adjusted for multiple testing using the Benjamini-Hochberg procedure [87]. Sequencing data were deposited in NCBI's Gene Expression Omnibus (GSE297002).

## Statistical analyses

GraphPad Prism 10 software (RRID: SCR_002798) was used for statistical comparison of live/dead cell counts, erythroid cell lineage percentage and immunoblot densitometry measurement. Student *t* test was used to assess significant

differences between groups. Visualization of RNAseq data was performed using EhancedVolcano (RRID:SCR_018931) and ComplexHeatmap (RRID:SCR_017270) in RStudio (R version 4.4.2) and adjusted in Inkscape (RRID:SCR_014479).

## Supporting information

**S1 Fig. Generation *Vps13a*/*Vps13c* double knockout mice. (A)** Schematic overview of the process used for the generation of *Vps13a* KO mice to be mated to previously generated *Vps13c* KO mice. *Vps13a*tm1a mice were first mated to Flp recombinase expressing mice to remove the LacZ and Neo cassettes to generate *Vps13a* conditional KO mice. Subsequently, mice were bred with β-actin-Cre expressing transgenic mice to remove exon 6 and generate constitutive full body *Vps13a* KO mice. **(B)** Anti-VPS13A western blots of lysates of cortical tissues from both WT and *Vps13a* KO mice confirming absence of VPS13A in the KO mice. **(C)** Genotyping results of adult mice and E12.5 embryos derived from *Vps13a*+/−/*Vps13c*−/− intercrosses. Values in the brackets indicate the number of expected mice/embryos based on mendelian distribution.
(TIFF)

**S2 Fig. (A)** Viability of cells derived from fetal livers assessed by trypan blue staining. Each dot represents one fetal liver with the specific genotype. **(B)** Percentage of each erythroid subset within the lineage negative gate. Each dot represents one fetal liver with the specific genotype. Results are presented as average plus SD of each subtype of erythroid cells. **(C)** The set of genes expressed by both WT and DKO S0 cells (*p* adj > 0.05 and |FC| < 1.2) was compared to the RNA-seq profile of different cell types reported by the "Descartes Cell Types and Tissues Database". The bar plot shows the top 10 cell types with transcriptomic profiles similar to those of S0 cells. **(D)** Heatmap showing normalized RNA count of key transcription factors that drives erythroid fate commitment and erythroid maturation in S0 cells. **(E)** Bar plot showing top 15 significantly upregulated (KEGG; top panel) and significantly downregulated (MSigDB; bottom panel) pathways based on adj P comparing DKO S0 cells versus WT. Raw data of this figure can be found at https://doi.org/10.5281/zenodo.15375803.
(TIFF)

**S1 Raw Images. Uncropped immunoblots.**
(PDF)

## Acknowledgments

We thank Yumei Wu, Chase Amos, Berrak Ugur, Hongyan Hao and Benjamin Johnson for discussion, Alina Vulpe for technical support, Timothy Nottoli (Yale Genome Editing Center) for the in vitro fertilization to generate *Vps13a* mutant mice, Ewa Menet (Yale Flow Cytometry Facility) for FACS support, Yale Center for Genome Analysis (YCGA) for RNAseq (supported by the National Institute of General Medical Sciences of the National Institutes of Health under Award Number 1S10OD030363-01A1), Francesc Lopez-Giraldez (Center for Biomedical Data Science, YCGA) for RNAseq analysis. We also thank Michael Schadt (Yale Department of Comparative Medicine) and Amos Brooks (Yale Department of Pathology) for their excellent histologic and immunohistochemical support.

## Author contributions

**Conceptualization:** Peng Xu, Diane S. Krause, Pietro De Camilli.

**Data curation:** Peng Xu, Rubia Isler Mancuso, Caroline J. Zeiss, Diane S. Krause, Pietro De Camilli.

**Formal analysis:** Peng Xu, Rubia Isler Mancuso, Caroline J. Zeiss, Diane S. Krause, Pietro De Camilli.

**Funding acquisition:** Diane S. Krause, Pietro De Camilli.

**Investigation:** Peng Xu, Rubia Isler Mancuso, Marianna Leonzino, Caroline J. Zeiss.

**Methodology:** Peng Xu, Rubia Isler Mancuso, Caroline J. Zeiss, Diane S. Krause, Pietro De Camilli.

**Project administration:** Diane S. Krause, Pietro De Camilli.

**Resources:** Peng Xu, Marianna Leonzino, Diane S. Krause, Pietro De Camilli.

**Supervision:** Peng Xu, Diane S. Krause, Pietro De Camilli.

**Validation:** Peng Xu, Rubia Isler Mancuso.

**Visualization:** Peng Xu, Rubia Isler Mancuso, Caroline J. Zeiss.

**Writing – original draft:** Peng Xu, Pietro De Camilli.

**Writing – review & editing:** Rubia Isler Mancuso, Marianna Leonzino, Caroline J. Zeiss, Diane S. Krause.

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
