## [Editor Report · Decision Letter 0]

16 May 2025

Dear Dr De Camilli,

Thank you for submitting your manuscript entitled "Defect in hematopoiesis and embryonic lethality at midgestation of Vps13a/Vps13c double knockout mice" for consideration as a Short Reports by PLOS Biology.

Your manuscript has now been evaluated by the PLOS Biology editorial staff as well as by an academic editor with relevant expertise and I am writing to let you know that we would like to send your submission out for external peer review.

Once your full submission is complete, your paper will undergo a series of checks in preparation for peer review. After your manuscript has passed the checks it will be sent out for review. To provide the metadata for your submission, please Login to Editorial Manager (https://www.editorialmanager.com/pbiology) within two working days, i.e. by May 20 2025 11:59PM.

Kind regards,

Ines

--

Ines Alvarez-Garcia, PhD

Senior Editor

PLOS Biology

---

## [Decision Letter · Decision Letter 1]

2 Jul 2025

Dear Dr De Camilli,

Thank you for your patience while your manuscript entitled "Defect in hematopoiesis and embryonic lethality at midgestation of Vps13a/Vps13c double knockout mice" went through peer-review at PLOS Biology. Your manuscript has now been evaluated by the PLOS Biology editors, an Academic Editor with relevant expertise, and by three independent reviewers.

As you will see, the reviewers find the conclusions interesting, but they also raise several issues that would need to be addressed before we can consider the manuscript for publication. Reviewer 1 raises several points that should be clarified, including whether or not the Vps13a and Vps13c embryonic mutants have a heartbeat or if there is evidence from the RNA-seq analysis that the cells are erythroid progenitors. Reviewer 2 notes that the cause of lethality is not established, and asks to comment on the fact that even if liver development is strongly affected, it does not mean that the activity of VSP13A and C is required in the liver, and also to improve the discussion on redundancy. Reviewer 3 thinks that the potential link between autophagy dysregulation and innate immune activation should be thoroughly discussed.

In light of the reviews, we are pleased to offer you the opportunity to address the comments from the reviewers in a revision that we anticipate should not take you very long. We will then assess your revised manuscript and your response to the reviewers' comments with our Academic Editor aiming to avoid further rounds of peer-review, although we might need to consult with the reviewers, depending on the nature of the revisions.

**IMPORTANT - SUBMITTING YOUR REVISION**

3. Resubmission Checklist

a) *PLOS Data Policy*

b) *Published Peer Review*

Sincerely,

Ines

--

Ines Alvarez-Garcia, PhD

Senior Editor

PLOS Biology

Reviewers' comments

Rev. 1:

In this paper the authors provide a brief report of data from double knockout, mouse embryos lacking the Vps13a and Vps13c genes and analyzed at a single timepoint (E12.5). They find that these embryos are small, pale, with a marked decrease in the size of the liver (figure 1A-E). Analysis of the fetal liver by flow cytometry with CD71 and Ter119 surface markers reveals a lack of normal erythroid differentiation with mutant cells almost entirely in the "S0" subpopulation (Figure 2). Comparison of RNA-Seq studies of normal and double knockout S0 cells revealed a down-regulation of genes associated with hematopoiesis and an up regulation of genes associated with interfere on signaling (Figure 3). The regulation of two of these genes (MDA5 and IFITM3) was also found in K562 cells lacking VPS13A and VPS13C (figure 4). In the discussion, the authors provide various speculations regarding the loss of there is lipid transfer genes and the regulation of genes associated with dsRNA sensing.

Overall, the data are novel, logically laid out, and clearly presented. However, there are a number of issues that bear consideration:

1. It is clear that the combined loss of Vps13a and Vps13c leads to embryonic lethality, however, it is not clear when that during embryogenesis they die. Did the E12.5 mouse have a heartbeat? (The timing of embryonic death may provide clues as to the etiology of the demise.)

2. The marked pallor at E12.5 indicates a severe defect in primitive erythropoiesis, since primitive red cells comprise nearly all of the circulating blood cells at this time point of mouse development. The flow cytometric analysis undertaken in the fetal liver indicates that definitive erythropoiesis is markedly blocked. Thus, the combined loss of these genes leads to profound defects in the differentiation of more than one lineage. This should be commented on and it may be useful to consider examining the circulating cells if there are enough to analyze in the mutant embryos.

3. It is not clear that the S0 comparison is one of "apples to apples". It is assumed that they represent erythroid progenitors but that may not be the case since this population basically lacks all surface markers analyzed. Is there evidence from the RNA-Seq analysis that these cells are indeed erythroid progenitors?

4. While the focus of this paper is a characterization of erythropoiesis using flow cytometry and RNA-Seq approaches in the fetal liver, the embryonic sections in figure 1 reveal marked defects in the central nervous system. It is surprising that there is no analysis or even mention of the CNS, given that several of the authors are in Neuroscience departments and that humans/Drosophila with VPS deficiencies have neurodevelopmental issues.

5. In the Discussion, the authors speculate that a causal relationship between (potential) organelle defects and the activation of innate immunity. Some evidence that there are indeed organelle defects in blood cells would considerably strengthen this conjecture. For example, are there defects in mitochondria/lysosomes/Golgi/ER in the S0 cells?

Rev. 2:

VPS13 proteins are a family of bridge-like lipid transport proteins (BLTPs) that mediate bulk lipid transfer at distinct subcellular contact sites. Among them, VPS13A and VPS13C are the most closely related and specific to vertebrates. In humans, loss-of-function mutations in either gene lead to distinct late-onset neurodegenerative diseases. However, their roles during early development have remained largely unclear. This study shows that while mice lacking either VPS13A or VPS13C individually are viable, double knockout (DKO) embryos die at midgestation. The authors identify impaired hematopoiesis and activation of innate immunity—specifically the RIG-I-mediated interferon response—as possible contributors to this lethality. Overall, the study provides valuable insights into the previously unrecognized functions of VPS13 in early mammalian development.

The main criticism is that the cause of lethality is not established. It would require tissue-specific KO or tissue specific rescue to ascertain that liver hematopoietic defects are causal and cell-autonomous to the lethality, which is beyond the scope of the present study. Since the hematopoiesis defect is presented with appropriately careful wording, this is not an issue, but perhaps the authors may want to comment on the fact that even if liver development is strongly affected, it does not mean that the activity of VSP13A and C is in fact required in the liver.

Similarly, the discussion is focused on the topic of redundancy. It is not clear from the data whether both proteins are indeed redundant, or whether the absence of either cause a mild stress, and the absence of both cause an additive and lethal one (with both stresses being more or less independent). That the group has localized both proteins to different contact sites might argue for the latter. Yet, given the mystery that shrouds the localization mechanisms for both proteins, it is also conceivable that, outside of highly transformed cultured cells, where the localization experiments have been done, the localization pattern of both proteins is much more overlapping than previously thought, consistent with a redundancy phenomenon.

At present the discussion offers a very short paragraph (the last one) to discuss these possibilities; a much larger one is dedicated to the cause of the lethality (erythropoiesis, cGAS-STING pathway), which, in the absence of tissue-specific KO and rescue, remains speculative.

Rev. 3:

This is an extremely well-conducted and clearly reported study that addresses the role of VPS13A and VPS13C in erythropoiesis.

The use of a double knockout (DKO) model convincingly demonstrates the functional redundancy of these lipid transport proteins and provides robust in vivo evidence for their critical involvement in red blood cell development and the regulation of innate immune responses. The authors support their conclusions with an extensive array of molecular, cellular, and hematological assays, offering a comprehensive characterization of the erythropoietic phenotype.

The manuscript is highly relevant and contributes important new insights into the connection between intracellular lipid transport, organelle homeostasis, and hematopoietic differentiation. Particularly noteworthy is the observation that disruption of inter-organelle lipid transport leads to erythroid-specific defects alongside aberrant activation of innate immune pathways.

However, one aspect remains insufficiently explored: the mechanistic link between VPS13A/C deficiency and the pronounced activation of interferon signaling specifically in erythroid cells. While the authors convincingly demonstrate induction of interferon-stimulated genes (ISGs) and involvement of RIG-I/MDA5 and cGAS-STING pathways, the upstream molecular events leading to this activation are not adequately discussed or experimentally addressed.

A plausible scenario worth further consideration involves the role of VPS13A and VPS13C in autophagy and mitophagy. These processes are essential for proper erythroid maturation, particularly in the clearance of mitochondria. Disruption of autophagy could lead to the accumulation of damaged mitochondria and leakage of mitochondrial DNA or double-stranded RNA into the cytosol—recognized triggers of cGAS-STING and RIG-I/MDA5 signaling, respectively (West et al., 2015; White et al., 2014; Sliter et al., 2018). VPS13A has been implicated in autophagy regulation, while VPS13C has a documented role in lysosomal lipid homeostasis. Defective mitophagic flux in erythroid precursors may therefore underlie the immune activation phenotype observed in this model.

Assessing whether the combined loss of VPS13A/C alters autophagy or mitophagy flux would substantially strengthen the study and offer a mechanistic explanation for the observed interferon response in erythroid cells.

In conclusion, this study represents a significant advancement in the field and opens new avenues for future research. Including or more fully discussing the potential link between autophagy dysregulation and innate immune activation would deepen the mechanistic insight and enhance the impact of the work.

---

## [Editor Report · Decision Letter 2]

8 Aug 2025

Dear Dr De Camilli,

Thank you for your patience while we considered your revised manuscript entitled "Defect in hematopoiesis and embryonic lethality at midgestation of Vps13a/Vps13c double knockout mice" for publication as a Short Report at PLOS Biology. This revised version of your manuscript has been evaluated by the PLOS Biology editors and the Academic Editor.

Based on our Academic Editor's assessment of your revision, we are likely to accept this manuscript for publication, provided you satisfactorily address the data and other policy-related requests stated below my signature.

In addition, we would like you to consider a suggestion to improve the title:

"VPS13A and VPS13C mediate partially redundant lipid transfer functions essential for hematopoiesis during murine embryogenesis"

We expect to receive your revised manuscript within two weeks.

*Published Peer Review History*

*Press*

Sincerely,

Ines

--

Ines Alvarez-Garcia, PhD

Senior Editor

PLOS Biology

DATA POLICY:

Many thanks for providing the data underlying the graphs shown in the figures. I have checked the data file in Zenodo and I am missing the data from Fig. 3B – a volcano plot. Please let me know where can I find the data or provide an updated version of the file containing it.

Likewise, I can't find the raw gel for Fig. S1B, thus please let me know where to find it or provide it if it is missing.

---

## [Editor Report · Decision Letter 3]

1 Sep 2025

Dear Dr De Camilli,

Thank you for the submission of your revised Short Report entitled "Impaired hematopoiesis and embryonic lethality at midgestation of mice lacking both lipid transfer proteins VPS13A and VPS13C" for publication in PLOS Biology. On behalf of my colleagues and the Academic Editor, Giovanni D'Angelo, I am delighted to let you know that we can in principle accept your manuscript for publication, provided you address any remaining formatting and reporting issues. These will be detailed in an email you should receive within 2-3 business days from our colleagues in the journal operations team; no action is required from you until then. Please note that we will not be able to formally accept your manuscript and schedule it for publication until you have completed any requested changes.

PRESS

Sincerely, 

Ines

--

Ines Alvarez-Garcia, PhD

Senior Editor

PLOS Biology
